# A polygenic risk score for the QT interval is an independent predictor of drug-induced QT prolongation

Steven T. Simon[1], Meng Lin[2], Katy E. Trinkley[3], Ryan Aleong[1], Nicholas Rafaels[2], Kristy R. Crooks[2], Michael J. Reiter[1], Christopher R. Gignoux[2], Michael A. Rosenberg[1]*

1 Division of Cardiology, University of Colorado School of Medicine, Aurora, CO, United States of America, 2 Colorado Center for Personalized Medicine, University of Colorado School of Medicine, Aurora, CO, United States of America, 3 Department of Clinical Pharmacy, School of Pharmacy, University of Colorado, Aurora, CO, United States of America

* michael.a.rosenberg@cuanschutz.edu

## Abstract

Drug-induced QT prolongation (diLQTS), and subsequent risk of torsade de pointes, is a major concern with use of many medications, including for non-cardiac conditions. The possibility that genetic risk, in the form of polygenic risk scores (PGS), could be integrated into prediction of risk of diLQTS has great potential, although it is unknown how genetic risk is related to clinical risk factors as might be applied in clinical decision-making. In this study, we examined the PGS for QT interval in 2500 subjects exposed to a known QT-prolonging drug on prolongation of the QT interval over 500ms on subsequent ECG using electronic health record data. We found that the normalized QT PGS was higher in cases than controls (0.212±0.954 vs. -0.0270±1.003, P = 0.0002), with an unadjusted odds ratio of 1.34 (95%CI 1.17–1.53, P<0.001) for association with diLQTS. When included with age and clinical predictors of QT prolongation, we found that the PGS for QT interval provided independent risk prediction for diLQTS, in which the interaction for high-risk diagnosis or with certain high-risk medications (amiodarone, sotalol, and dofetilide) was not significant, indicating that genetic risk did not modify the effect of other risk factors on risk of diLQTS. We found that a high-risk cutoff (QT PGS ≥ 2 standard deviations above mean), but not a low-risk cutoff, was associated with risk of diLQTS after adjustment for clinical factors, and provided one method of integration based on the decision-tree framework. In conclusion, we found that PGS for QT interval is an independent predictor of diLQTS, but that in contrast to existing theories about repolarization reserve as a mechanism of increasing risk, the effect is independent of other clinical risk factors. More work is needed for external validation in clinical decision-making, as well as defining the mechanism through which genes that increase QT interval are associated with risk of diLQTS.

**Data Availability Statement:** Data cannot be shared publicly because the Colorado Multiple Internal Review Board informed consent process employed by the Colorado Center for Personalized

Medicine Biobank did not include an option for public sharing of data, nor did the COMIRB approval for retrospective analysis of electronic health record data. Data are available from the University of Colorado Institutional Data Access/ Ethics Committee (contact via website form: ucdenver.edu/offices/institutional-research-and-effectiveness/datarequest or email at IR@UCDenver.edu) for researchers who meet criteria for access to confidential data.

**Funding:** This work was supported by the National Heart, Lung, and Blood Institute of the National Institutes of Health (MAR: R01HL146824), as well as Google, Inc. (MAR). The funders had no role in study design, data collection and analysis, decision to publish, or preparation of the manuscript.

**Competing interests:** The authors have declared that no competing interests exist.

## Introduction

Drug-induced long QT syndrome (diLQTS) is the precursor to the potentially fatal arrhythmia, torsades de pointes (TdP), and despite its rarity, has major clinical ramifications due to the association with both cardiac [1] and noncardiac medications [2–4]. Accurate prediction of diLQTS prior to administration of a medication would have a high impact on management decisions. There might be alternatives that could be selected if an individual were identified as having an increased risk of diLQTS [5]; for example, changing to an antibiotic that is not associated with diLQTS. If no alternative medications can be found, a patient could be referred for closer monitoring [6] or engage in a shared decision-making discussion about the risks and benefits of use. To date, a number of investigations have proposed models to predict diLQTS based on clinical [7–10] and genetic [11, 12] information; however, methods to integrate these prediction models into clinical care decision pathways remain elusive.

To date, there have been no genome-wide association studies (GWAS) specifically targeting the phenotype of diLQTS; however, a number of GWAS have been completed with accurate genetic prediction of the resting QT interval [13–18]. The potential biological link between resting QT interval and diLQTS comes via a principle called repolarization reserve [19–21], which is based on the idea that the redundancy of the ion channels responsible for repolarization is compromised in certain individuals based on genetic variability (i.e., mutations) or clinical conditions, such as heart failure or electrolyte disturbances, such that certain individuals are more likely to develop diLQTS with exposure to known culprit medications than others. This principle is supported by the finding that 5–10% of people who develop TdP after drug exposure harbor rare mutations associated with long-QT syndrome [22–24], and forms the basis for guidelines recommending avoidance of known QT-prolonging medications for individuals with congenital LQTS [25]. In prior work, a significant association between a polygenic risk score (PGS) for the QT interval and risk of TdP [11] was observed, as well as a correlation in the degree of QT prolongation in healthy volunteers [11]. However, these studies did not specifically consider repolarization reserve reflecting the interaction of genetic risk and specific medications associated with diLQTS, and in smaller populations, we had previously failed to detect a signal of interaction between polygenic risk and QT prolongation [12].

The need for better prediction models for diLQTS is evident from inspection of current clinical decision-support (CDS) tools that have been developed and implemented in hospital systems to alert providers to potential risk [26, 27]. When evaluated after development, the impact of these CDS systems has been modest at best [28], with most studies noting a change in prescriber orders for medications [8, 29]. We recently examined the impact of a CDS based on the prior QT interval alone and noted that there was a paradoxical increase in mortality in those patients in whom the provider acknowledged the alert compared with those who ignored it [30]. This work suggests that more accurate prediction models for diLQTS could improve long-term results of CDS tools.

As such, an investigation of the potential role of genetic prediction for models to prevent diLQTS can examine both the biological effect of genes associated with baseline QT prolongation with exposure to known QT-prolonging medications (i.e., the repolarization reserve hypothesis), as well as the various ways in which genetic risk can be incorporated into risk-prediction models for diLQTS. In this investigation, we seek to examine both, as we first examine the potential interaction of genetically determined QT interval, via the polygenic risk score, and then explore the linear and nonlinear impact of genetic risk on risk of diLQTS.

**Table 1. Baseline characteristics.**

|  | Cases (N = 281) | Controls (N = 2219) | Total (N = 2500) | P-value (Case versus control) |
|---|---|---|---|---|
| Average age (± SD) | 56.5 ± 15.8 | 52.8 ± 15.9 | 53.2 ± 15.9 | 0.0002 |
| Female sex (%) | 150 (53.4%) | 1247 (56.2%) | 1397 (55.9%) | 0.370 |
| Hispanic ethnicity (%) | 22 (7.8%) | 252 (11.4%) | 274 (11.0%) | 0.075 |
| Caucasian/White race (%) | 219 (77.9%) | 1723 (77.7%) | 1942 (77.7%) | 0.951 |
| African-American/Black race (%) | 27 (9.6%) | 226 (10.2%) | 253 (10.1%) |  |
| Other/unknown race (%) | 35 (12.5%) | 270 (12.2%) | 305 (12.2%) |  |
| AF diagnosis (%) | 107 (38.1%) | 300 (13.5%) | 407 (16.3%) | < 0.001 |
| HF diagnosis (%) | 115 (40.9%) | 281 (12.7%) | 396 (15.8%) | < 0.001 |

SD = Standard deviation; AF = Atrial fibrillation diagnosis; HF = Heart failure diagnosis

## Results

Among 2500 participants in the Colorado Center for Personalized Medicine Biobank with documented exposure to known QT-prolonging medications, there were 281 cases of diLQTS (11.2%) and 2219 controls (88.8%), Table 1. Among 20 cardiovascular traits, the PGS for QT interval had the strongest association with diLQTS (S1 Table in S1 File), and was the only trait with a significant unadjusted association after adjusting for multiple comparisons. PGS for systolic blood pressure (P = 0.024), diastolic blood pressure (P = 0.027), high-density lipoprotein (P = 0.0053) and apolipoprotein A level (P = 0.012) were noted to be closest after QT interval, although these PGS were not explored further.

For the normalized QT interval PGS (mean ~0, standard deviation = 1), the range was -3.856 to 3.236, inter-quartile range (IQR) was -0.651 to 0.701. Across all medications and patients, the QT interval PGS was higher in cases than controls: 0.212±0.954 vs. -0.0270±1.003, P = 0.0002 (Fig 1), with an unadjusted odds ratio of 1.277 (95%CI 1.124–1.450, P < 0.001) for

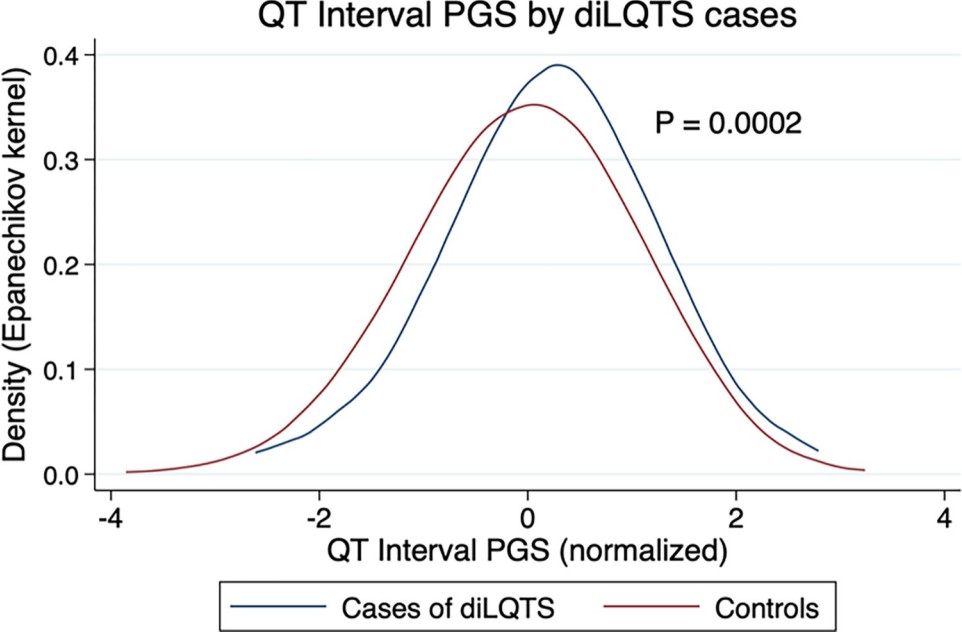

**Fig 1. QT PGS by cases (diLQTS) and controls.** Kernel density estimates, Epanechikov kernel function.

**Table 2. Risk of diLQTS by medication.**

| Drug | OR$_{PGS}$ (95%CI) | OR$_{Drug}$ (95%CI) | P-value$_{Interaction}$ | N |
|---|---|---|---|---|
| Amiodarone | 1.28 (1.11–1.47)* | 5.80 (3.87–8.67)** | 0.513 | 118 |
| Levofloxacin | 1.26 (1.10–1.44)* | 1.79 (1.20–2.66)* | 0.588 | 201 |
| Ondansetron | 1.45 (1.20–1.76)** | 0.70 (0.54–0.91)* | 0.076 | 1630 |
| Ciprofloxacin | 1.28 (1.12–1.45)** | 1.01 (0.51–2.00) | 0.916 | 93 |
| Erythromycin | 1.26 (1.11–1.44)** | 1.06 (0.33–3.39) | 0.157 | 39 |
| Fluconazole | 1.27 (1.12–1.46)** | 1.49 (0.96–2.32) | 0.806 | 171 |
| Haloperidol | 1.27 (1.11–1.45)** | 0.99 (0.61–1.61) | 0.776 | 190 |
| Moxifloxacin | 1.29 (1.13–1.46)** | 0.68 (0.11–4.14) | 0.180 | 24 |
| Azithromycin | 1.27 (1.11–1.45)** | 0.83 (0.44–1.58) | 0.647 | 126 |
| Citalopram | 1.29 (1.13–1.47)** | 1.58 (0.91–2.73) | 0.675 | 100 |
| Dofetilide | 1.34 (1.17–1.54)** | 8.84 (5.07–15.41)** | 0.923 | 63 |
| Methadone | 1.30 (1.14–1.47)** | 1.62 (0.60–4.41) | 0.051 | 33 |
| Propofol | 1.29 (1.13–1.47)** | 2.93 (1.89–4.55)** | 0.783 | 118 |
| Sotalol | 1.27 (1.12–1.45)** | 1.07 (0.35–3.29) | 0.576 | 33 |

*Indicates P < 0.01

** Indicates P < 0.001

association with diLQTS. Self-assigned sex, ethnicity, and race were not significantly associated with diLQTS (Table 1), while increased age was associated with diLQTS (OR 1.02 per year, 95%CI 1.01–1.02, P < 0.001). After adjustment for age, sex, race, and ethnicity, the association of QT interval PGS and diLQTS remained significant with odds ratio of 1.306 (95%CI 1.148–1.486, P < 0.001). Interestingly, the interaction of the QT PGS in African-American subjects was inverse to what it is in Caucasians (S1 Fig in S1 File). This interaction was statistically significant, with a P value for the interaction term of 0.003, and an odds ratio of 0.454 (95%CI 0.271–0.760). On exploration of possible mechanisms, we noted that African-Americans were younger than Caucasians (46.4 ± 0.97 years old versus 54.9 ± 0.35 years old for Caucasians, P < 0.0001), and had a lower probability of AF diagnosis (OR 0.42, 95%CI 0.27–0.67, P < 0.001). There was no significant difference in use of high-risk medications, including amiodarone, levofloxacin, or propofol, nor an increased risk of HF diagnosis (S2 Table in S1 File). The distribution of QT PGS was also different in African-Americans than Caucasians (S2 Fig in S1 File), with a bimodal distribution in the African-American population.

When the association with diLQTS was examined by QT-prolonging medication (Table 2), we found that the QT PGS was independently associated with risk of diLQTS after adjustment for each type of medication, and that only certain medications were independently associated with diLQTS; specifically, amiodarone, levofloxacin, dofetilide, and propofol were associated with increased risk and ondansetron was associated with decreased risk, a finding we have previously reported in this dataset [10] (Table 2). However, we also found that for none of the medications was there evidence of a significant interaction between the QT PGS and the medication (all interaction term P-values > 0.05, Table 2).

To explore the role of patient-level prediction using other diagnostic codes, we calculated the MIC across 8315 diagnostic and procedure codes (S3 Table in S1 File), from which we identified the diagnoses of heart failure (HF) and atrial fibrillation (AF) to be associated with higher risk of diLQTS based on MIC values, and for which variables were created to indicate the presence of one or both of these diagnoses (N for HF = 396 (15.84%) and for AF = 407 (16.28%)). For patients with a HF diagnosis, the risk of diLQTS was OR 4.78 (95%CI 3.65–

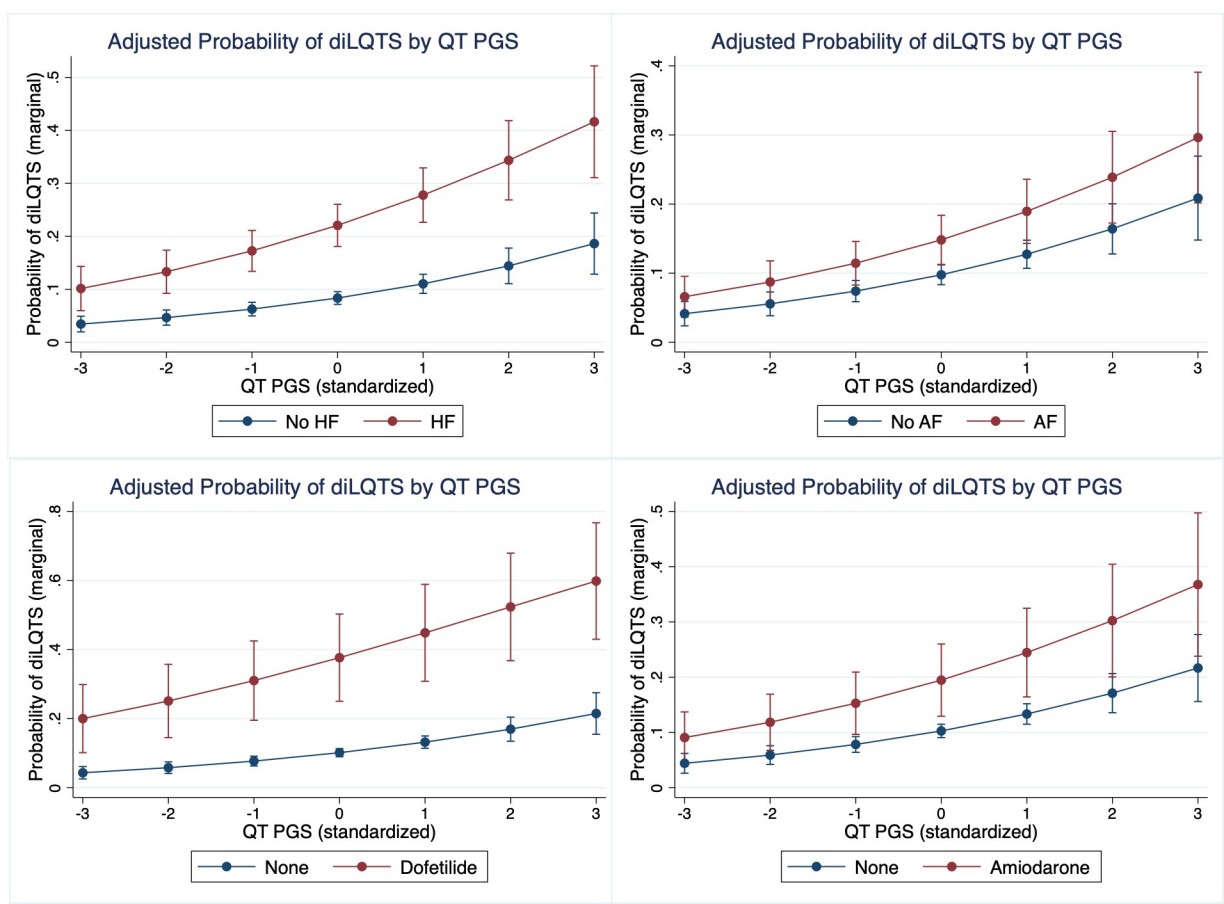

**Fig 2. Marginal (adjusted) probability of diLQTS by level of QT PGS for predictors.** A. Heart failure (HF), B. Atrial fibrillation (AF), C. Dofetilide, D. Amiodarone.

6.25, P < 0.001) and for AF the OR was 3.93 (95%CI 3.00–5.15, P < 0.001) in unadjusted models. Note that in models with AF and HF diagnosis included, the association of age with diLQTS was no longer significant (OR 1.00 per year, 95%CI 0.99–1.01, P = 0.836). In a model with age, both diagnoses, and PGS, the OR for diLQTS was OR 1.34, 95%CI 1.17–1.53, P<0.001 (Age OR 1.00, 95%CI 0.99–1.01, P = 0.667; HF OR 3.70, 95%CI 2.78–4.92, P < 0.001; AF OR 2.92, 95%CI 2.13–3.99, P < 0.001 in multivariable models). For neither diagnosis nor age was there evidence of an interaction with PGS (HF interaction P-value = 0.698; AF interaction P-value 0.407, age interaction P-value = 0.813). Age and AF/HF diagnoses were then combined with specific medications that had a significant association with increased risk of diLQTS (amiodarone, dofetilide, levofloxacin, and propofol; Table 1) and QT PGS to examine adjusted risk of diLQTS (Fig 2). (Note: ondansetron was dropped from this multivariate model due to P value 0.967 after adjustment for other medications; others remained significant at P < 0.05—largest P value was levofloxacin at P = 0.013).

To examine nonlinear effects of PGS, we first examined polynomials (k = 3) of QT PGS in unadjusted and adjusted models, in which we found that no polynomial for QT PGS was significant in either unadjusted or adjusted models, with only linear terms having a significant association. We then examined quantiles of QT PGS (Table 3) and restricted cubic splines of QT PGS in adjusted (Fig 3A and 3B) and unadjusted models (Fig 3C and 3D). Neither quantiles nor spline models performed better than the linear model based on information criteria

**Table 3.**

| 3A. Quintiles | | | 3B. Spline knots | |
| --- | --- | --- | --- | --- |
| Quintile | Min | Max | Knots | QT PGS |
| 1 | -3.86 | -0.82 | 1 | -1.70 |
| 2 | -0.82 | -0.22 | 2 | -0.57 |
| 3 | -0.22 | 0.26 | 3 | 0.02 |
| 4 | 0.26 | 0.88 | 4 | 0.62 |
| 5 | 0.88 | 3.24 | 5 | 1.54 |

or C-statistic. (Linear: AUC = 0.751, AIC 1538.3, BIC 1590.7, df = 9; Quantile: AUC = 0.744, AIC 1545.0, BIC 1614.8, df = 12; Spline: AUC = 0.750, AIC 1541.4, BIC 1611.3, df = 12).

Since prior work [31] had proposed use of cutoffs to identify high- or low-risk patients based on 'genetic risk', we explored the two possible clinical applications of PGS cutoffs—a lower cutoff ($\leq 2$ standard deviations below mean) to rule-out lower risk subjects and a higher cutoff ($\geq 2$ standard deviations above mean) to rule-in higher risk subjects. Based on these cutoffs, 69 subjects (2.76%) were considered low-risk for diLQTS. After adjustment for medications, AF and HF, the low-risk cutoff was not a significant predictor of diLQTS (OR 0.44, 95% CI 0.17–1.15, P = 0.094), with 6 (8.7%) low-risk individuals actually having diLQTS events.

For the high-risk cutoff, after adjustment for age, medications, HF and AF, there was a statistically significant increase in risk of diLQTS (OR 3.48, 95%CI 1.61–7.52, P = 0.002) among

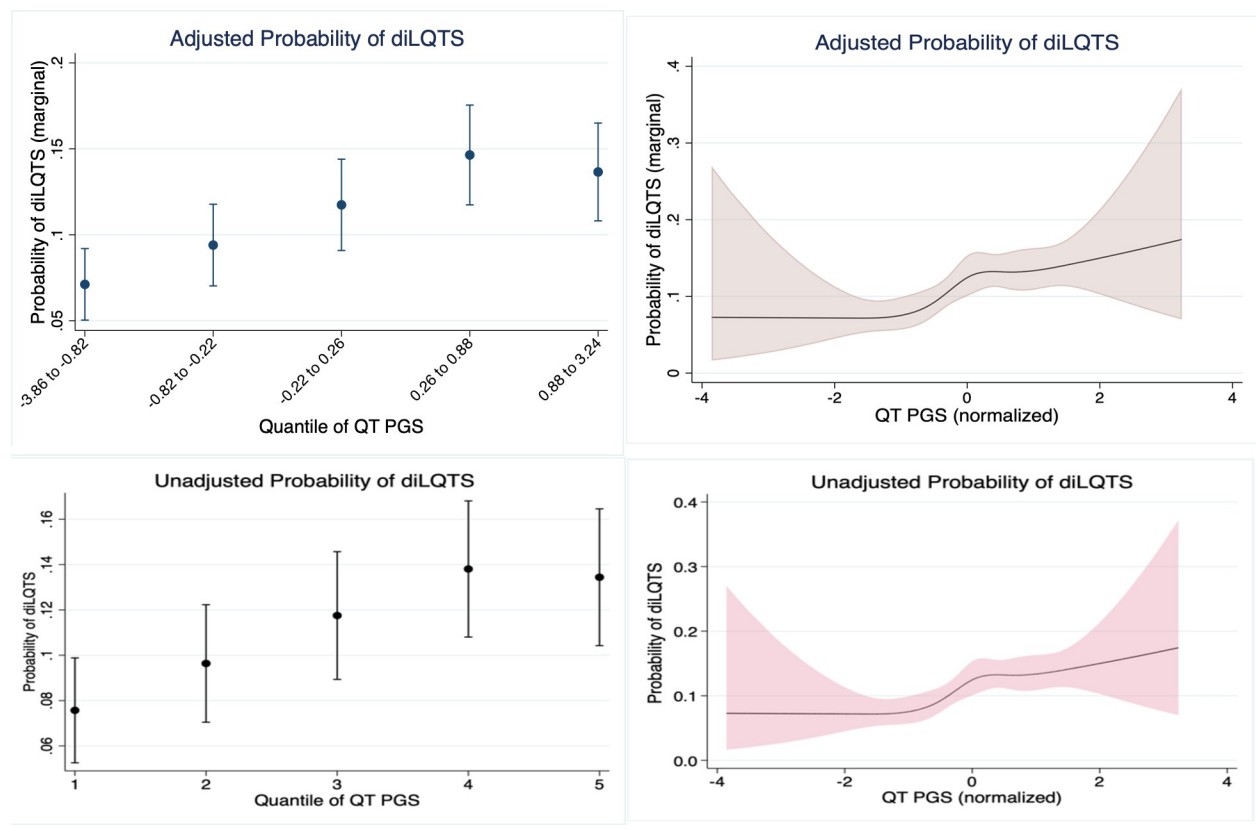

**Fig 3. Nonlinear evaluation of QT PGS.** A. Adjusted quintiles, B. Adjusted restricted cubic splines, C. Unadjusted quintiles, D. Unadjusted restricted cubic splines.

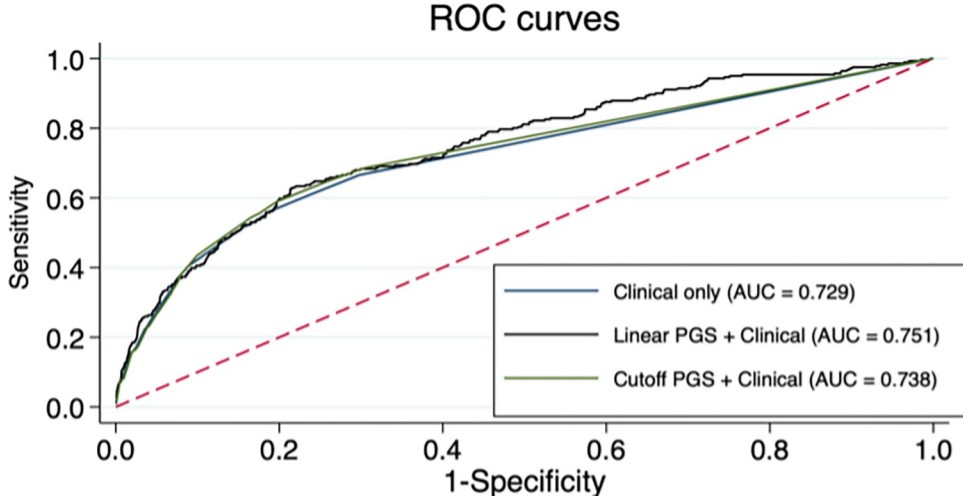

**Fig 4. ROC curves for various models predicting diLQTS.**

those in the highest risk category. Among these individuals (N = 39) there were 11 who had diLQTS (28.2%), and 28 (71.8%) who did not, compared with 270 (11.0%) of individuals who were not high-risk based on QT PGS. As a comparison, the model based solely on clinical predictors had 34 subjects classified as high-risk (based on predicted probability threshold over 50%), of whom 21 (61.8%) had diLQTS. In the combined model with QT PGS as a linear predictor, 51 subjects were predicted to be at risk (probability ≥ 50%), of whom 31 (60.8%) had diLQTS. Comparison of area-under-ROC curve (AUC, Fig 4) showed that the linear PGS model with clinical predictors (AUC = 0.751) was statistically superior to the clinical predictors only (AUC = 0.729, P = 0.04) and comparable to the model with high-risk cutoff combined with clinical predictors (AUC = 0.738, P = 0.20).

To explore methods to integrate QT PGS into clinical decision-making, we examined creation of a decision tree, in which we found that nodes based on presence of a HF diagnosis, an antiarrhythmic agent (amiodarone, sotalol, or dofetilide), and a high-risk QT PGS (≥ 2 standard deviations above the mean) provided some measure of discrimination of risk (Fig 5), although we did not have an external dataset in which to evaluate the decision-tree model.

## Discussion

In this single-center study of PGS for prediction of drug-induced QT prolongation, we found that a model based on genetic predictors of QT interval (QT PGS) provided independent risk prediction for diLQTS, after adjustment for medications and other clinical risk factors, including age, HF and AF, noting that sex, race, and ethnicity were not significant predictors of diLQTS in our population, and that the effect of age was no longer significant after adjustment for HF and AF. We found that the contribution of genetic predictors to a model of standard clinical factors improved predictive ability in an independent manner, such that there did not appear to be an association with either high-risk diagnoses or with certain high-risk medications, specifically the antiarrhythmic drugs amiodarone, sotalol, and dofetilide. While we did not detect any significant nonlinear impact of the polygenic score for QT prolongation across the range of scores, we noted that a cutoff of QT PGS ≥ 2 standard deviations above the mean was able to provide some level of increased risk prediction above clinical predictors, which we demonstrated with integration into a decision-tree, as might be applied in real-world decision-support tools within the electronic health record.

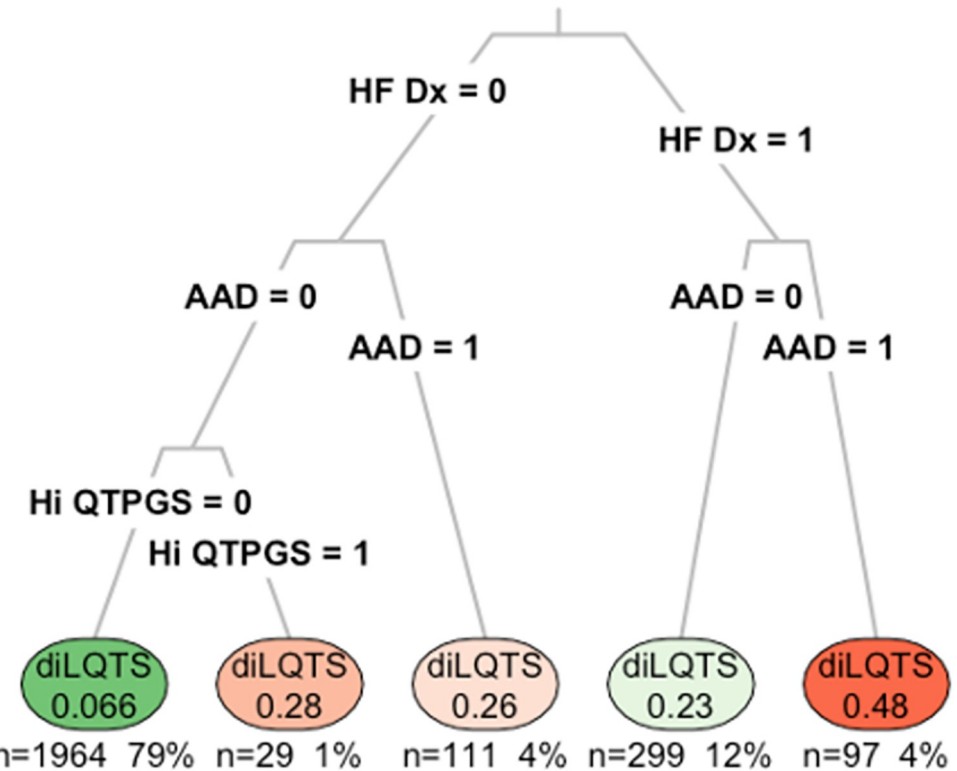

**Fig 5. Decision-tree example for integration of QT PGS.** Entry into the model is based on a patient for whom a known QT-prolonging medication is to be prescribed. Nodes corresponding to any heart failure diagnosis (HF Dx), use of an Class III antiarrhythmic agent (AAD), and QT PGS $\geq$ 2 SD above the mean are used to define the decision process. Listed within the leaves (bottom) are the proportion of subjects with diLQTS, and the coverage as a number and percentage of the total population.

The promise of clinical applications of PGS across diseases is that they provide a possible biomarker that can be used to guide clinical practice. Specific to diLQTS, such information could be easily integrated into clinical decision-support tools, particularly in the emerging era of biobanks such as available at the University of Colorado [32, 33], incorporating genetic information that could be obtained prior to other clinical diagnostic information, such as the QT interval itself on an ECG [26, 30]. However, to understand how this information might be used clinically, more work is needed to examine the predictive accuracy itself, as well as how the information might be integrated into care decisions. In previous work on this dataset, we examined the use of machine-learning methods [9, 10] that integrated large amounts of clinical data in order to make a prediction of risk for diLQTS. One of the challenges with use of such 'big data' methods is the need for integration within the EHR, since application of a machine-learning method would require some method to maintain and retrain models, and with it concerns about compatibility with the electronic framework or methods to map inputs from diagnostic codes, medications, and procedure codes into the model to generate the prediction. As a contrasting approach, many EHR systems, including the one at the University of Colorado [26, 27, 30], maintain an active system of rule-based alerts for which simple criteria can be easily integrated in the clinical care process, such as those applied in decision trees. The possibility that results from a study can be directly examined within such a rule-based system thus

provides an alternative to more sophisticated methods, albeit at the trade-off of prediction accuracy—a situation we have previously described for diLQTS [10].

In this manner, when addressing a new possible biomarker such as a PGS, one can follow one of two roads towards clinical implementation. The first is to identify methods to integrate the raw data directly, either using a multivariate prediction model, or perhaps a machine-learning model, and with it identify a method to ensure the model can be run on the existing framework and maps inputs appropriately. The second is to identify cutoffs or dichotomize inputs such that the model can be integrated into a rule-based decision tree, as we have outlined here, in which we selected cutoffs from which to derive those at either increased or decreased risk of diLQTS. There is precedence in this approach, for example in the application of the biomarker B-type natriuretic peptide (BNP), which is associated with heart failure exacerbation [34], and was noted to have a direct clinical application in the emergency department [35], where values below a selected cutoff of 100 pg/mL [36] were used to 'rule out' heart failure exacerbation as a mechanism of shortness of breath. In theory, such an approach could be applied in use of QT PGS for diLQTS, where a cutoff could be used to identify subjects who reach a sufficiently high or low risk as to warrant more stringent (higher risk) or relaxation (lower risk) of monitoring. In this study, we found the higher cutoff to be predictive in this respect, but not the lower; however, it cannot be overstated that these results are hypothesis-generating, and would require external validation to confirm predictive accuracy.

One interesting observation from our investigation was that there did not appear to be any evidence of an interaction between medications or clinical diagnoses (AF or HF) and genetic prediction of the QT interval (QT PGS), as might suggest a biological impact of genetics on risk of diLQTS. This finding is in contrast to the theory of repolarization reserve [37], which suggested that certain patients with lower repolarization reserve (i.e., higher QT PGS) might be more susceptible to diLQTS in the presence of high-risk medications or diagnoses. While we did not have additional clinical factors, such as serum potassium or magnesium levels, which could provide additional support to our findings, the fact that we were unable to obtain any signal of impact suggests that this conceptual model may not be entirely applicable in an in-patient clinical setting when genes are combined in the form of a PGS. In addition, we did not have a comparison group of subjects who were not exposed to known QT-prolonging medications, which would be needed to confirm this finding as it could be possible that any QT-prolonging medication of any effect size could potentiate genetic risk. As such, more work is needed to evaluate the hypothesis of repolarization reserve in other settings, such as in out-patients or in those unexposed to known QT-prolonging medications.

The other interesting observations concerned the role of demographic information, including age, sex, and race, with regard to the risk of diLQTS. For one, we did not detect an association of female sex with risk of diLQTS, which is in contrast with a number of prior studies that have noted this association [38–40], although not all [41]. Additionally, while we noted that increased age was associated with the risk of diLQTS, this association appeared to be modulated through increased risk of AF and HF, both of which were a significant risk factor of diLQTS. When included with these factors in a multivariable model, the effect of age was no longer significant. Other studies have noted the association of age and diLQTS [40, 42], although the challenge of disentangling concominant diseases and treatments, including class III anti-arrhythmic agents such as amiodarone and dofetilide, perhaps limits what can be extrapolated from existing literature.

We have previously examined the effect of PGS for resting QT interval across subjects of non-Caucasian ancestry, where we found the PGS derived in Caucasians was not an accurate predictor of resting QT interval, nor that the PGS created from populations of African ancestry reached the same level of predictive accuracy as the Caucasian-derived score (likely due to

inadequate power/sample size) [12]. In this study, we noted an intriguing inverse association of the Caucasian-derived PGS for QT interval in subjects of African ancestry and risk of diLQTS. Despite efforts at exploring potential mechanisms of this finding, we could not uncover a straightforward explanation for this association. Further work will be needed to better-understand the role of ancestry and genetic risk-prediction for diLQTS.

Our study was limited in several respects, a number of which have been discussed previously in analysis of this population [9, 10]. For one, there are concerns about noise that is inherent in use of diagnostic codes and electronic health record data in general; specific to our study, we did not prospectively obtain ECG tracings in all patients exposed to known QT-prolonging medications, but had only data from those who had an ECG performed for clinical purposes. This selection bias is likely to have led to potentially fewer diagnoses of diLQTS, and also highlights concerns that the association with diagnoses such as HF and AF may have resulted from these individuals having an ECG performed as part of work-up for those diagnoses. Only through systematic collection of ECGs and QT measurements can one overcome such a bias, and it is unknown how this bias would have impacted the role of genetics in prediction of diLQTS.

A second limitation of our findings was that the PGS applied for diLQTS was based on the resting QT interval, and not specifically the entity of diLQTS. As discussed above, the rationale of this application is based on the biological hypothesis of 'repolarization reserve', which we were unable to validate through examination of drug-PGS and disease-PGS interactions, and rather noted an inverse association with African-American race and risk of diLQTS with increasing QT PGS. To our knowledge, there has not been a PGS developed specifically for diLQTS that has been validated to the same extent as the PGS for resting QT interval, and it is possible that such a score could provide a better method for risk-stratification than the methods proposed here. Future work on genetic risk of diLQTS in the future will be needed to examine this possibility.

In conclusion, we found that genetic risk of QT prolongation, in the form of a polygenic risk score, was an independent predictor of diLQTS, and that while linear models of polygenic risk provided superior predictive accuracy, use of a cutoff $\geq 2$ standard deviations above the mean may provide a method for more facile clinical implementation. We could not confirm prior hypotheses about the role of repolarization reserve as a mechanism of genetic risk of diLQTS [11], although further work is needed to examine this theory directly. More work is also needed on external validation and clinical integration to confirm the application of PGS in clinical decision-making.

## Materials and methods

### Study population

The study population has been described previously [9, 10]. Briefly, the broader study population was obtained through query of the electronic health record (EHR) to obtain 35,639 inpatients treated between 2003 and 2018 with at least 1 of 39 medications associated with risk of diLQTS, who had an electrocardiogram in the system performed within 24 hours of medication administration. The primary outcome of diLQTS was defined as any individual with an ECG that had a computer-measured QTc over 500ms at any point during the inpatient encounter after exposure to a known QT-prolonging agent, after exclusion of ECGs with conduction disease (QRS $\geq 120$ ms due to bundle branch block, intraventricular conduction disease, ventricular pacing). Predictors included any medication or diagnosis (ICD-9 or ICD-10) listed in the medical record that was present at the time of medication administration, which totaled 8315 diagnostic codes. From this original dataset, we examined 2500 subjects with

genotype data available as part of the Colorado Center for Personalized Medicine (CCPM) Biobank, from which calculation of polygenic risk score (PGS) for QT interval and other cardiovascular traits could be determined. Among these 2500 subjects, there were 14 known QT-prolonging medications to which they were exposed (Table 1).

## Polygenic risk scores

The CCPM Biobank (CCPM) is an institutional biobank with broad consent from subjects for genetic analyses and EHR review. Ethical approval and consent were reviewed and approved by the Colorado Multiple Institutional Review Board (#15–0461). Genomic DNA was extracted from whole blood (Potassium/sodium ethylenediaminetetraacetic acid tubes) samples, and genotyping was performed on a custom single-nucleotide polymorphism (SNP) microarray (MultiEthnic Global Array, Illumina, Inc). Microarray data were subjected to a per-sample 0.985 call-rate filter. The genetic information was further imputed through the Trans-Omics for Precision Medicine (TOPMed) imputation server [43], and annotated as Freeze-2 of the CCPM Biobank, with loci having $r^2 > 0.7$ being kept (For detailed information, see Wiley et al. [33]). The PGS for QT interval and other cardiac phenotypes (S1 Table in S1 File) was calculated based on weights obtained from the largest genome-wide association study (GWAS) for QT interval to date [13], and normalized across the biobank population. The PGS weight files were retrieved from the PGS catalog [44] (https://www.pgscatalog.org). Quality control of the weights was performed before any score calculation, including (1) removing the variants with ambiguous allele codes to avoid strand mismatch with the target data (i.e. the subjects in CCPM freeze 2), (2) removing variants with an allele code that is different from those in the target data, despite strand flipping, (3) and other standard process such as matching genome builds and variant IDs with those in the target, discarding duplicates, and matching strands (full description of the pipeline is available at www.menglin44. com/ESCALATOR). The score was calculated under the standard definition of PGS using PLINK 2.0 [45] (Chang et al., GigaScience, 2015), as additive of weight values across variants weighted by genotypes for each individual. Each score was normalized by subtracting the mean and dividing by the standard deviation across the genotyped population.

## Analysis plan

We first examined the correlation of normalized PGS for each of the 20 cardiac traits with the label of diLQTS using a standard one-way analysis of variance (ANOVA) comparison, with P-values adjusted for multiple testing according to Bonferroni correction (alpha = 0.05/20 = 0.0025). Because only the QT PGS reached a level of statistical significance based on this criterion (S1 Table in S1 File), we focused the rest of the analysis on this score. We first modeled unadjusted risk of diLQTS for QT PGS using logistic regression, reporting the odds ratio (OR), followed by adjusted models for each of the 14 known QT-prolonging medications available in this dataset, including interaction terms to examine effect modification. To identify associated diagnostic codes, we performed maximum entropy coefficient (MIC; minepy. MIC, version 1.2.6), which examines both linear and non-linear associations with diLQTS based on mutual information [46], and ranked diagnoses based on association with diLQTS, from which we categorized the most associated diagnoses to examine in models. We then examined adjusted models that included specific highly associated medications and diagnostic categories along with QT PGS. To examine for nonlinearity and possible cutoffs for use in clinical prediction, we performed logistic regression using quantiles and restricted cubic splines. We then examined clinical decision applications through examination of high ($\geq 2$ standard deviations above the mean) and low ($\leq 2$ standard deviations below the mean) QT PGS, and

compared models with only clinical predictions and linear QT PGS. Models were compared based on Akaike and Bayes' information criteria, as well as area-under ROC curve ('roccomp' module in Stata), with specific attention on categorization of risk across models. Finally, we performed decision tree analysis using the R package (rpart, version 4.1.19). Results are presented as estimates from logistic regression models, as well as marginal predictions from fit models. Data processing and cleaning was performed using Python (version 2.9.12). Analysis was performed using Stata IC, version 17 (StataCorp, Inc., College Station, TX, USA). The authors did not have access to information that could be used to identify individual participants during or after data collection.

## Supporting information

**S1 File.**
(DOCX)

## Acknowledgments

We would also like to thank Michelle Edelmann and Premanand Tiwari of the University of Colorado Health Data Compass for assistance with data acquisition.

## Author Contributions

**Conceptualization:** Steven T. Simon, Katy E. Trinkley, Ryan Aleong, Nicholas Rafaels, Christopher R. Gignoux, Michael A. Rosenberg.

**Data curation:** Meng Lin, Nicholas Rafaels, Kristy R. Crooks, Christopher R. Gignoux, Michael A. Rosenberg.

**Formal analysis:** Meng Lin, Michael A. Rosenberg.

**Funding acquisition:** Michael A. Rosenberg.

**Investigation:** Steven T. Simon, Meng Lin, Ryan Aleong, Nicholas Rafaels, Michael J. Reiter, Christopher R. Gignoux, Michael A. Rosenberg.

**Methodology:** Steven T. Simon, Meng Lin, Katy E. Trinkley, Ryan Aleong, Kristy R. Crooks, Michael J. Reiter, Christopher R. Gignoux, Michael A. Rosenberg.

**Project administration:** Kristy R. Crooks, Christopher R. Gignoux, Michael A. Rosenberg.

**Resources:** Nicholas Rafaels, Kristy R. Crooks, Christopher R. Gignoux, Michael A. Rosenberg.

**Software:** Meng Lin, Nicholas Rafaels, Michael A. Rosenberg.

**Supervision:** Christopher R. Gignoux, Michael A. Rosenberg.

**Validation:** Steven T. Simon, Katy E. Trinkley, Ryan Aleong, Kristy R. Crooks, Michael J. Reiter, Michael A. Rosenberg.

**Visualization:** Michael A. Rosenberg.

**Writing – original draft:** Meng Lin, Michael A. Rosenberg.

**Writing – review & editing:** Steven T. Simon, Meng Lin, Katy E. Trinkley, Ryan Aleong, Nicholas Rafaels, Kristy R. Crooks, Michael J. Reiter, Christopher R. Gignoux, Michael A. Rosenberg.

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
