## [Decision Letter · Decision Letter 0]

13 Feb 2024

PONE-D-23-17422A Polygenic Risk Score for the QT Interval is an Independent Predictor of Drug-Induced QT ProlongationPLOS ONE

Dear Dr. Rosenberg,

Thank you for submitting your manuscript to PLOS ONE. After careful consideration, we feel that it has merit but does not fully meet PLOS ONE’s publication criteria as it currently stands. Therefore, we invite you to submit a revised version of the manuscript that addresses the points raised during the review process.

We look forward to receiving your revised manuscript.

Kind regards,

Katriina Aalto-Setala, Professor

Academic Editor

PLOS ONE

Journal Requirements:

Additional Editor Comments:

There are several comments and criticism raised by the reviewers.  Please, check them and reply accordingly and mark clearly the changes made in the manuscript.

Reviewers' comments:

Reviewer's Responses to Questions

**Comments to the Author**

1. Is the manuscript technically sound, and do the data support the conclusions?

Reviewer #1: Yes

Reviewer #2: Partly

2. Has the statistical analysis been performed appropriately and rigorously? 

Reviewer #1: Yes

Reviewer #2: No

3. Have the authors made all data underlying the findings in their manuscript fully available?

Reviewer #1: No

Reviewer #2: Yes

4. Is the manuscript presented in an intelligible fashion and written in standard English?

Reviewer #1: Yes

Reviewer #2: Yes

5. Review Comments to the Author

Reviewer #1: This manuscript is well written and the analysis is sound, although I think it could be further improved by some comments provided below:

- The introduction could benefit from including a final paragraph on which the authors state what are the objectives of this work in a synthesized and clear manner.

- on the results section, there is a list of drugs that have been tested or associated with risk of increased QTc interval, but it is not clear why these drugs were analyzed or how the patients were selected and if they were given alone or in combination. This needs to be stated in a clear way in the manuscript, even more so when the results are before methods. More information about drugs given or its own or together that could cause drug-drug interactions is also needed.

- In addition, a description of the diseases for which these specific drugs are used and their distribution in the population could be good.

- the lack of a control group without exposure to QT prolonging agents is a strong limitation, I think it would need further discussion of what this implies in terms of the results obtained, and that it should be added in the list of limitations

- A bit more discussion on what the limitations imply is needed. Would the AUC ROC curves be improved if you had more information on the patients? Would it be worth presenting a small subset of patients of which you might have all this information that is known to affect QT? Would the genetic markers remain significant if you had this information for the patients?

Reviewer #2: This is an interesting study of the effects of polygenic QT interval risk prediction as an independent predictor of drug induced LQTS. Greater than 2 standard deviations above the mean polygenic risk score is suggested as a clinically relevant threshold for precision therapeutic discussion making. The paper does not include external validation of the observed effects and does not directly test the effect of using PGS in clinical decision-making, yet is an interesting contribution to the literature and suggests future studies to validate the findings and trial use of PGS in a clinical setting may be warranted. I have a few comments and concerns that I have listed below.

My main comment is that QT interval is known to be differential by sex, age, and race yet these key variables were ignored in the analyses. This could confound the results in important ways, for example, if QT prolonging drugs are differentially prescribed in women and men, and if women are at greater risk of developing heart failure given their elevated baseline QT intervals. Similarly, older people may be at greater risk of diLQT because of their elevated QT intervals compared to younger people, which may result in differential prescription rates, and confounding in the models. While I agree that the PGS (based on autosomal variants) may not change due to sex or age, the effect of the PGS on risk of diLQT very well may be different by age and sex. Differential performance of the PRS, which was developed a European cohort, is a major potential confounder given that African Americans are at twice the OR for drug-induced QT prolongation compared to others. These variables was not explored because the authors claim that the data were not available, but this information should be readily available in the EHR data and sex at birth and ancestry at least can certainly be derived from the genetic data, so it's not clear why these important measures were left out of the analysis.

Minor:

Please include a standard "table 1" with demographic information about the sample including sex, age, race/ethnicity/ancestry, HF, AF, etc.

Please include a histogram of the PRS scores.

"Among 2500 subjects with genotype data, there were 281 cases of diLQTS (11.2%) and 2219 controls (88.8%)." Some additional description in this paragraph would help- e.g. "Among 2500 participants of the Colorado biobank taking QT-prolonging medication, there were 281 cases..."

The authors claim that the linear PGS model with clinical predictors (AUC = 0.751) was superior to the clinical predictors only (AUC = 0.722), but it's not clear if these are significant differences.

The PGS values were standardized in Fig 2, but it's not clear to me how they were standardized?

6. PLOS authors have the option to publish the peer review history of their article (what does this mean?). If published, this will include your full peer review and any attached files.

Reviewer #1: No

Reviewer #2: No

---

## [Author Response · Author response to Decision Letter 0]

27 Mar 2024

We thank the reviewers for their helpful and insightful comments, and in the response below have tried to respond as best as possible. Original comments are in bold and responses are in italics. 

Reviewer #1: This manuscript is well written and the analysis is sound, although I think it could be further improved by some comments provided below:

- The introduction could benefit from including a final paragraph on which the authors state what are the objectives of this work in a synthesized and clear manner.

-- We thank the reviewer for this helpful suggestion, and have added this paragraph as requested in the Introduction. As we note, this study actually examines the role of genetic risk of diLQTS from two directions: the biological, in which we examine the repolarization reserve hypothesis directly with regard to drug-genetic interactions (noting that we could not confirm this hypothesis in our population for any drug); and the clinical, in which we examined the role of genetic risk as a linear and nonlinear predictor of diLQTS for use in clinical prediction algorithms and models. 

- on the results section, there is a list of drugs that have been tested or associated with risk of increased QTc interval, but it is not clear why these drugs were analyzed or how the patients were selected and if they were given alone or in combination. This needs to be stated in a clear way in the manuscript, even more so when the results are before methods. More information about drugs given or its own or together that could cause drug-drug interactions is also needed.

-- We thank the reviewer for this note, for which we have included a brief description of why the drugs were included, noting that the drugs selected were identified from standard databases of medications with a high-risk association with diLQTS. We should note here, and in response to additional comments below, that we have examined the association of these medications, alone or in combination with others, in prior work that was not limited to only subjects who had provided genetic samples for genotyping (see Simon et al., 2021 and 2022, references #9 and #10). Note that the full dataset, not limited to those subjects in whom genotypes could be obtained, was over 35K subjects, in contrast with the 2500 in whom we had genotype data for this study. While we acknowledge that such drug-drug interaction information would be valuable in exploration of association with genetic risk, it must be noted that 1) we had concerns that the inadequate sample sizes would have a high probability of inferring false-negative association of a combination of known QT-prolonging in certain patients, and 2) that in this study the fact that we failed to detect any significant interaction of individual medications with genetic risk (the subject of this investigation) would suggest that further exploration of gene-drug interactions for combinations of medications would also be unlikely to produce findings capable of withstanding the scrutiny of concerns about data mining or fitting noise rather than signal. 

- In addition, a description of the diseases for which these specific drugs are used and their distribution in the population could be good.

-- Unfortunately, we did not have specific drug-indication data available in our database, and having performed additional database extraction work in follow-up of this study, would suggest that the actual ‘indication’ of a given medication often cannot be easily obtained without detailed query of each subject’s chart and review of notes (an impossibility given the de-identified nature of this data). For example, amiodarone is a common medication used for both atrial and ventricular arrhythmias, although determination of which arrhythmia cannot just be determined from review of the problem list, as patients with atrial fibrillation might be treated for ventricular tachycardia, or vice versa. As such, we might infer that each medication was used for the common indications (e.g., levofloxacin antibiotic for infection), but cannot state beyond this point whether the indication was appropriate, whether the treating provider was aware of a possible risk of diLQTS, and whether alternative agents were considered. 

- the lack of a control group without exposure to QT prolonging agents is a strong limitation, I think it would need further discussion of what this implies in terms of the results obtained, and that it should be added in the list of limitations

-- We agree that this analysis of observational data is limited, as is any study of observational data by confounding, bias, and the inherent inability to establish causal inference regarding whether the use of a specific agent actually ‘caused’ diLQTS or was merely present when other causative agents were present. However, we disagree that the proper control for this study would have been a group without exposure to a known QT-prolonging agent, since the focus on this investigation was prolongation of QT interval after exposure to medications (i.e., diLQTS). De novo QT prolongation in the absence of offending agents, and congenital long-QT syndromes are certainly of clinical interest, and an important topic of exploration for future investigations, although are probably beyond the scope of this study. 

- A bit more discussion on what the limitations imply is needed. Would the AUC ROC curves be improved if you had more information on the patients? Would it be worth presenting a small subset of patients of which you might have all this information that is known to affect QT? Would the genetic markers remain significant if you had this information for the patients?

 -- Unfortunately, we only have the dataset available; as noted above, since this study employed de-identified data, more granular exploration of patient-level factors that could improve predictive accuracy is thus beyond the scope of this investigation. As such, it is unknown whether ‘stronger’ clinical predictors could improve the present models or not, and we are hesitant to speculate on what sort of information could lead to a better model, with or without genetic predictors. However, we agree with the reviewer that the limitations should include an important consideration with regard to the capability of the PGS examined, which is that the genetic risk score for QT interval that was applied was specific for resting QT interval—a genetic marker or score that was associated with diLQTS itself, or for specific medications, could potentially improve predictive accuracy, which we have discussed in the revised manuscript on page 13 of the discussion. 

Reviewer #2: This is an interesting study of the effects of polygenic QT interval risk prediction as an independent predictor of drug induced LQTS. Greater than 2 standard deviations above the mean polygenic risk score is suggested as a clinically relevant threshold for precision therapeutic discussion making. The paper does not include external validation of the observed effects and does not directly test the effect of using PGS in clinical decision-making, yet is an interesting contribution to the literature and suggests future studies to validate the findings and trial use of PGS in a clinical setting may be warranted. I have a few comments and concerns that I have listed below.

My main comment is that QT interval is known to be differential by sex, age, and race yet these key variables were ignored in the analyses. This could confound the results in important ways, for example, if QT prolonging drugs are differentially prescribed in women and men, and if women are at greater risk of developing heart failure given their elevated baseline QT intervals. Similarly, older people may be at greater risk of diLQT because of their elevated QT intervals compared to younger people, which may result in differential prescription rates, and confounding in the models. While I agree that the PGS (based on autosomal variants) may not change due to sex or age, the effect of the PGS on risk of diLQT very well may be different by age and sex. Differential performance of the PRS, which was developed a European cohort, is a major potential confounder given that African Americans are at twice the OR for drug-induced QT prolongation compared to others. These variables was not explored because the authors claim that the data were not available, but this information should be readily available in the EHR data and sex at birth and ancestry at least can certainly be derived from the genetic data, so it's not clear why these important measures were left out of the analysis.

-- We greatly thank the reviewer for this important comment, and as suggested, have obtained baseline demographic information for the cohort examined in this investigation, including age, sex, race, and ethnicity. Interestingly, and perhaps surprisingly, the only demographic factor that was associated with diLQTS in our cohort was age, which itself was only significant without inclusion of the diagnosis of atrial fibrillation or heart failure. In other words, it would seem that the increased risk of diLQTS associated with increased age was primarily as a risk factor for these conditions, since inclusion of either or both diagnoses negated the effect of age. We were somewhat surprised to observe that sex was not a significant predictor, since female sex itself is a risk factor for QT prolongation without drug exposure. Nonetheless, we have addressed these results in the Results and Discussion of the revised manuscript on pages 5-6 and 12-13.

The reviewer raises a critical issue with regard to the role of race/ethnicity in genetic prediction of diLQTS, which our group has previously noted in application of genetic risk scores for resting QT interval (see Rosenberg et al., reference #12), where the PGS was predictive in individuals of Caucasian but not African-American ancestry. In this study, we found that race and ethnicity were not independent risk factors for diLQTS in unadjusted or adjusted models. However, when we examined the interaction of the genetic risk score for QT interval with risk of diLQTS, we noted a fascinating observation that the impact of the QT PGS in African-American subjects was inverse to what it is in Caucasians (New Supplemental Figure 1). This interaction was statistically significant, with a P value for the interaction term of 0.003, and an odds ratio of 0.454 (95%CI 0.271 – 0.760). On exploration of possible mechanisms, we noted that African-Americans were younger than Caucasians (46.4 ± 0.97 years old versus 54.9 ± 0.35 years old for Caucasians, P < 0.0001), and had a lower probability of AF diagnosis (OR 0.42, 95%CI 0.27 – 0.67, P < 0.001), although neither association alone seemed to explain this finding. There was no significant difference in use of high-risk medications, including amiodarone, levofloxacin, or propofol, nor an increased risk of HF diagnosis (New Supplemental Table 3). The distribution of QT PGS was also very different in African-Americans than Caucasians (Supplemental Figure 2), with a bimodal distribution in the African-American population. There are a number of possible explanations for the different distributions, ranging from mis-application of race (a self-applied demographic in the EHR in our health system) or a higher prevalence of mixed-Ancestry subjects in the African American population, although we withheld speculation due to lack of an ability to test these hypotheses. We have highlighted these new findings in the revised manuscript on page 5 (Results), and page 13 (Discussion). 

While these factors are certainly a subject of future investigations, we note that when applied across the population as a whole, the impact on primary results was minimal, likely due to the significantly larger number of individuals of Caucasian subjects in the population. To the reviewer’s point, future prospective evaluation of the high cut-off model proposed will be important, as will the additional evaluation of PGS-based and other clinical prediction models in non-Caucasian populations. 

Minor:

Please include a standard "table 1" with demographic information about the sample including sex, age, race/ethnicity/ancestry, HF, AF, etc.

-- We have included this Table in the revised manuscript as requested.

Please include a histogram of the PRS scores.

¬-- We have included race-based density plots in the revised manuscript, as indicated above, as well as the existing Figure 1, which displays kernel densities of the distribution of PGS scores by case. We believe these plots demonstrate more meaningful information with regard to the impact of race and diLQTS status than an overall histogram, but can provide one if requested (see below for example). 

"Among 2500 subjects with genotype data, there were 281 cases of diLQTS (11.2%) and 2219 controls (88.8%)." Some additional description in this paragraph would help- e.g. "Among 2500 participants of the Colorado biobank taking QT-prolonging medication, there were 281 cases..."

-- We have included this information in the revised manuscript on page 4 as requested. 

The authors claim that the linear PGS model with clinical predictors (AUC = 0.751) was superior to the clinical predictors only (AUC = 0.722), but it's not clear if these are significant differences.

-- The linear model was statistically better (P=0.04) than the clinical predictors only model, but not statistically different than the high-risk model (P=0.20). We have added these comparisons to the revised manuscript on page 9. 

The PGS values were standardized in Fig 2, but it's not clear to me how they were standardized?

-- All PGS values were normalized by subtracting the mean and dividing by the standard deviation. We have included this information in the revised manuscript on page 16.

---

## [Decision Letter · Decision Letter 1]

23 Apr 2024

A Polygenic Risk Score for the QT Interval is an Independent Predictor of Drug-Induced QT Prolongation

PONE-D-23-17422R1

Dear Dr. Rosenberg,

We’re pleased to inform you that your manuscript has been judged scientifically suitable for publication and will be formally accepted for publication once it meets all outstanding technical requirements.

Kind regards,

Katriina Aalto-Setala, Professor

Academic Editor

PLOS ONE

Additional Editor Comments (optional):

Reviewers' comments:

Reviewer's Responses to Questions

**Comments to the Author**

1. If the authors have adequately addressed your comments raised in a previous round of review and you feel that this manuscript is now acceptable for publication, you may indicate that here to bypass the “Comments to the Author” section, enter your conflict of interest statement in the “Confidential to Editor” section, and submit your "Accept" recommendation.

Reviewer #2: All comments have been addressed

2. Is the manuscript technically sound, and do the data support the conclusions?

Reviewer #2: Yes

3. Has the statistical analysis been performed appropriately and rigorously? 

Reviewer #2: Yes

4. Have the authors made all data underlying the findings in their manuscript fully available?

Reviewer #2: Yes

5. Is the manuscript presented in an intelligible fashion and written in standard English?

Reviewer #2: Yes

6. Review Comments to the Author

Reviewer #2: The authors present a revised manuscript on the effects of polygenic QT interval risk prediction as an independent predictor of drug induced LQTS. The authors have address my comments and the paper is improved, thanks!

7. PLOS authors have the option to publish the peer review history of their article (what does this mean?). If published, this will include your full peer review and any attached files.

Reviewer #2: No

---

## [Editor Report · Acceptance letter]

4 May 2024

PONE-D-23-17422R1 

PLOS ONE

Dear Dr. Rosenberg, 

I'm pleased to inform you that your manuscript has been deemed suitable for publication in PLOS ONE. Congratulations! Your manuscript is now being handed over to our production team.

Kind regards, 

on behalf of

Dr Katriina Aalto-Setala 

Academic Editor

PLOS ONE